# Health and economic costs of early and delayed suppression and the unmitigated spread of COVID-19: The case of Australia

Tom Kompas[1]*, R. Quentin Grafton[2], Tuong Nhu Che[3], Long Chu[2], James Camac[4]

**1** Centre of Excellence for Biosecurity Risk Analysis, School of Biosciences and School of Ecosystem and Forest Sciences, University of Melbourne, Melbourne, Australia, **2** Crawford School of Public Policy, Australian National University, Canberra, Australia, **3** Australian Centre for Biosecurity and Environmental Economics, Crawford School of Public Policy, Australian National University, Canberra, Australia, **4** Centre of Excellence for Biosecurity Risk Analysis, School of Biosciences, University of Melbourne, Melbourne, Australia

* tom.kompas@unimelb.edu.au

**Data Availability Statement:** Data Availability: All data and accompanying MATLAB model code can be obtained at: Kompas, T., Grafton, Q., Che, N., Chu, L., & Camac, J. (2020, December 5). COVID-

## Abstract

We compare the health and economic costs of early and delayed mandated suppression and the unmitigated spread of 'first-wave' COVID-19 infections in Australia in 2020. Using a fit-for-purpose SIQRM-compartment model for susceptible, infected, quarantined, recovered and mortalities on active cases, that we fitted from recorded data, a value of a statistical life year (VSLY) and an age-adjusted value of statistical life (A-VSL), we find that the economic costs of unmitigated suppression are multiples more than for early mandated suppression. We also find that using an equivalent VSLY welfare loss from fatalities to estimated GDP losses, drawn from survey data and our own estimates of the impact of suppression measures on the economy, means that for early suppression not to be the preferred strategy requires that Australia would have to incur more than 12,500–30,000 deaths, depending on the fatality rate with unmitigated spread, to the economy costs of early mandated suppression. We also find that early rather than delayed mandated suppression imposes much lower economy and health costs and conclude that in high-income countries, like Australia, a 'go early, go hard' strategy to suppress COVID-19 results in the lowest estimated public health and economy costs.

## 1. Introduction

As of the end of May 2020, the global number of COVID-19 cumulative cases and reported fatalities, respectively, exceeded 6 million and 370,000 [1]. In response to the pandemic, many countries have imposed various types of suppression measures at different times to reduce the growth rate in COVID-19 infections [1]. Severe suppression measures (e.g., school and university closures, travel restrictions, mandatory social distancing requirements), when implemented in combination, have received the moniker, 'lock-down'. Some have questioned whether the 'cure' (i.e., the lock-down) may be more costly in terms of the economy than no

19 Australia. Retrieved from osf.io/mn89p. We have also included the 'Full Data' set Excel file as a Supporting information file.

**Funding:** The author(s) received no specific funding for this work.

**Competing interests:** The authors have declared that no competing interests exist.

or limited suppression measures (e.g., [2–4]). This question can only be resolved with an epidemiological model of COVID-19 infections estimated from actual data combined with an empirical economic model of the costs of a lock-down. Using a combined epi-economic modelling approach we estimate and compare the public health and economic costs of early and late suppression measures as well as the impacts of unmitigated spread.

We evaluate whether a lock-down to suppress COVID-19 in Australia was justified by estimating the health and economic costs (including welfare costs of COVID-19 patients) for three scenarios. The first scenario is mandated 'early suppression' and is what actually occurred with the imposition of a lock-down that began in March 2020, and that we simulate continued until the end of May 2020. The second is mandated 'delayed suppression' or the effect of imposing a lock-down either 14, 21 or 28 days after it was actually imposed. For completeness, the third scenario is an 'unmitigated spread' counter-factual that assumes there were no Australian government (and voluntary) suppression measures. For each scenario, we estimate the number of fatalities, hospitalisations, and direct economy costs.

To obtain results we use a **S**usceptible, **I**nfected, **Q**uarantined, **R**ecovered, and **M**ortality (SIQRM) compartment model we constructed for this purpose. Estimates of the welfare losses for COVID-19 patients are calculated using an Australian value of statistical life year (VSLY) and an age-adjusted value of a statistical life (A-VSL). Our estimates of the effects on the economy are drawn from established survey data and our own estimates of the impacts from losses on tourism and the effects of the extent or strictness of suppression measures on the economy.

## 2. COVID-19 in Australia and policy responses

COVID-19 was first detected in Australia in January 25[th], 2020, from a passenger who arrived in Melbourne from Wuhan on January 19[th]. As of May 30[th], 2020, there were 7,173 reported cases and of which 6,582 have recovered with 102 deaths [5]. Fig 1 provides a summary of cumulative reported infected cases and the daily reported growth rate (3-day average) during this period.

The first recorded death from COVID-19 in Australia was on March 1[st], 2020, when the total number of cases in the country was 29. During the initial stages of the infection, the total number of cases (including overseas arrivals) and the number of active cases grew exponentially, which prompted the Australian government to respond with a series of sequential and increasingly strict control measures.

Fig 2 summarizes the dynamics of COVID-19 and the main control responses in Australia since the first Australian death. Control responses varied by state and territory. From March 15[th], the Australian federal government required international arrivals to self-quarantine for 14 days. Three days later, the government banned non-essential gatherings and encouraged social distancing (i.e., 1-5m distance and 4m$^2$ per person). On March 21[st], the federal government ordered non-essential businesses to close. Non-essential businesses included pubs, licensed clubs, hotels, places of worship, gyms, indoor sporting venue, cinemas, and casinos. Shortly thereafter, further tightening occurred with more restrictions on funeral and wedding attendance, fitness classes, and arcades [6].

The most severe mandated suppression measures were introduced on March 30[th] when public gatherings were reduced to a maximum of two people. From this date, Australians were required to stay at home unless shopping for essentials, receiving medical care, limited (30 minute) exercising, or traveling to and from work or for educational purposes. All suppression measures combined 'flattened the curve' and rapidly reduced the number of reported infected cases. Fig 2 shows that recoveries started exceeding new infections on April 4[th] 2020 when the number of recorded active cases peaked at 4,935.

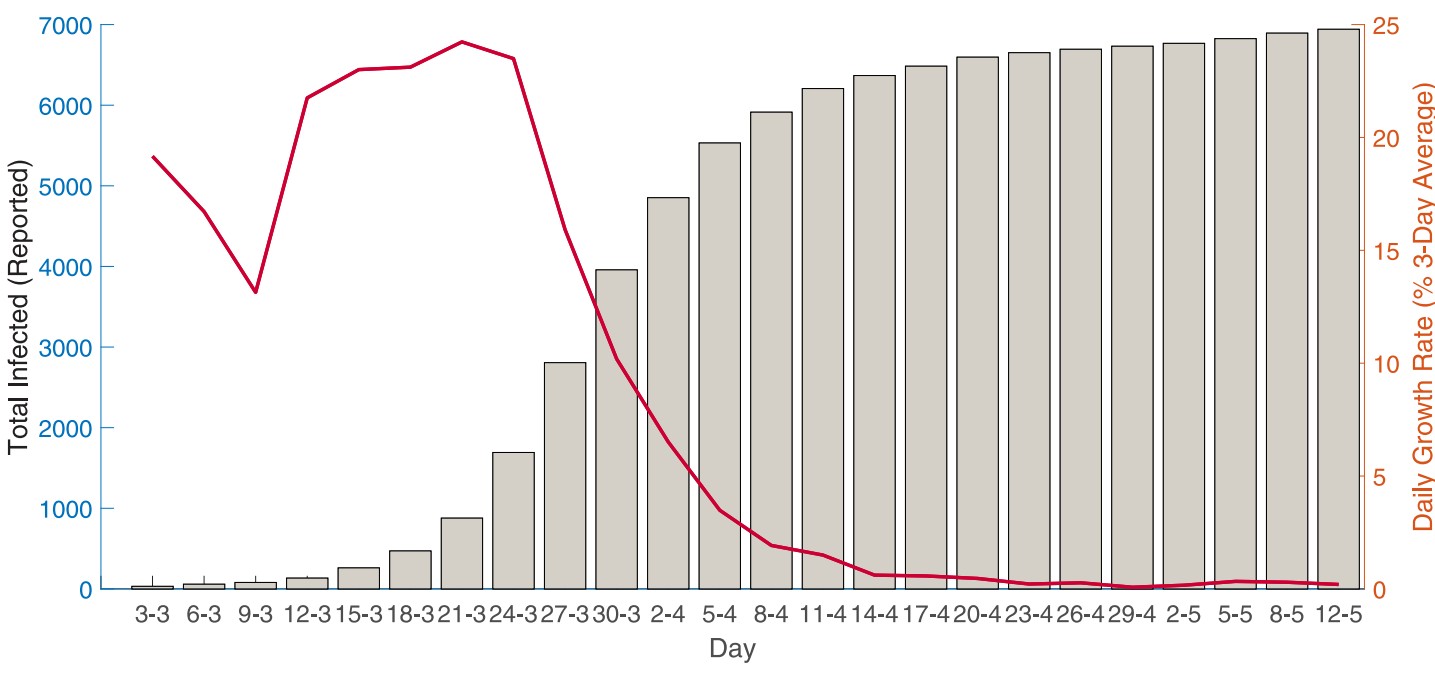

**Fig 1. COVID-19 infected cases and 3-day average growth rate.**

As of 1st June 2020, Australian mandated suppression measures drastically reduced community transmission of the virus and the daily growth rate (the daily increase in the total number of cases over the total number of cases, on a three-day average) declined from around 25%, with 268 new recorded cases on 22nd March, to a daily growth rate of 0.26% and 11 new recorded cases on 30th May 2020 [5].

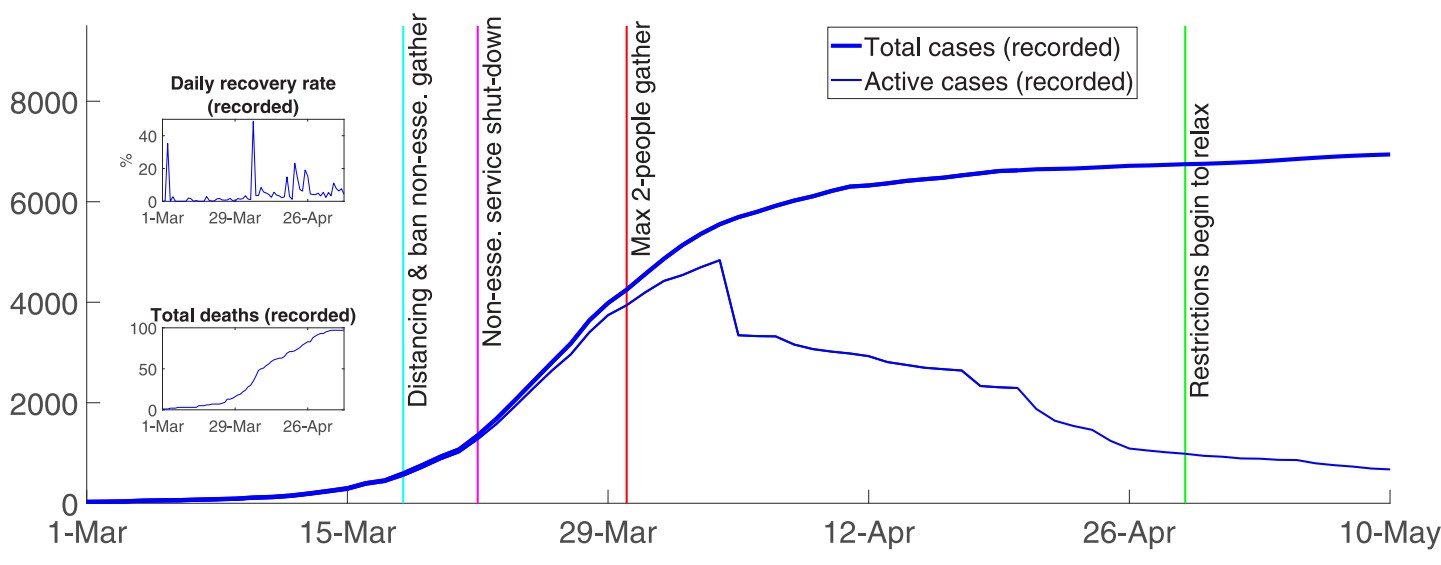

**Fig 2. Overview of COVID-19 in Australia.**

## 3. Material and methods

### 3.1. Epidemiological model

We constructed an epidemiological model of COVID-19 infections in Australia that has five compartments, **S**usceptible, **I**nfected, **Q**uarantined, **R**ecovered, and **M**ortality, using recorded data for **Q**, **R**, and **M**. Susceptible people can be infected with the virus via community transmission and become infectious. People who are infected can also come from abroad and transmit the disease before clinical signs appear. We assumed, as implemented in Australia, that all those who are officially reported to be infected are 'quarantined', recorded as active cases, in hospital or self-isolating, and either recover or die. Suppression measures mitigate the spread of the infection while the speed of recovery depends on treatment protocols.

Our model is formalized as follows:

$$\frac{dS}{dt} = -\frac{R_0}{T^i}[1-\epsilon] \times I \frac{S}{S+I+Q+R} + \mu^b(S+I+Q+R) - \mu^d S \tag{1}$$

$$\frac{dI}{dt} = \frac{R_0}{T^i}[1-\epsilon] \times I \frac{S}{S+I+Q+R} - \mu^d I - \frac{1}{T^i}I + W \tag{2}$$

$$\frac{dQ}{dt} = \frac{1}{T^i}I - \chi Q - fQ - \mu^d Q \tag{3}$$

$$\frac{dR}{dt} = \chi Q - \mu^d R \tag{4}$$

$$\frac{dM}{dt} = fQ \tag{5}$$

where $S$, $I$, $Q$, $R$, $M$ are the number of susceptible, infected, recorded active cases (i.e., quarantined), and eventually recorded as recovered and deceased, all functions of time; $W$ is the number of infected people who arrive from abroad; $R_0$ is the basic reproduction rate (see Wu et al. [7]); $\epsilon$ is the effectiveness of suppression measures; $T^i$ is the infectious period; $\mu^d$ is the natural mortality rate (we estimate $\mu^d = 0.7\%$ using the mortality data from the Australian Institute of Health and Welfare [8] and population data from ABS [9]); $\mu^b$ is the birth rate which we specify $\mu^b = 1.3\%\backslash$ using the estimates from ABS [10]; $\chi$ is the rate at which active cases recover, which is observable (as in the top subplot in Fig 2); and $f$ is the rate at which infected people die because of the virus, where $f = 1.67\%$, as given from reported data in late April (noting that reported COVID-19 cases underestimated the true population infection rate (see Phipps et al. [11]), and varies daily over the model run from March 1st to April 20th [5]. Economy and cost impacts are also shown for the fatality rate in Verity et al. [12] of 0.7%.

The policy variable $\epsilon(t)$ represents mandated suppression measures and is delimited by $0 \leq [1-\epsilon(t)] \leq 1$, for combined mandated and voluntary measures. Thus, without suppression measures $[1-\epsilon(t)]$ approaches one (i.e., $\epsilon(t)$ approaches 0) and with the most restrictive suppression measures $[1-\epsilon(t)]$ approaches zero (i.e., $\epsilon(t)$ approaches 1).

The total number of susceptible people is assumed to be 70% of the total population. From the SIQRM model in Eqs 1–5, the total number of recorded cases ($T$) are calculated as the sum of active cases (in quarantine), recovered, and deaths, or $T = Q + R + M$, all of which were observed.

It is important to note that the model, although fit for purpose, is an abstraction of the relevant infection stages and does not include any explicit time lag between infection and infectiousness, which has impact on the practical timing of the predicted epidemic in Australia.

## 3.2. Model parameters and forecasting

We estimated parameters and tested the model to actual recorded data from March 1st to April 20th, 2020 and test the model on recorded data thereafter. There are five parameters to estimate. The first is the infectious period ($T^i$), the period over which an individual can spread the virus (which can occur before developing symptoms) to ceasing to spread the virus because of quarantine, recovery, or death. The second is the basic reproduction number ($R_0$). The other three parameters include the reduction in community transmission after Australian governments introduced each of the three principal suppression measures, i.e., March 19th -21st, March 22nd–28th, and after March 28th, all lagged before coming into effect by the appropriate number of days (see Fig 2).

We fitted the number of recorded cases projected by the model to the number of observed cases using a non-linear least squares technique [13] that estimated the parameters by minimizing the sum of the squared distance between the projected and actual values in Eq (6):

$$\hat{\beta}\left(y_t^o, X_t^o\right) = \mathrm{arg}min_{<\beta>} \sum_{t=0}^{T^o} \left(y_t^o - SIQRM_t\left(\beta|X_t^o\right)\right)^2 \qquad (6)$$

where $y_t^o$ is the observed total number of cases (i.e., recorded cases) at time $t$, which differs from the actual number of infected people; $X_t^o$ is other observable information at time $t$ (e.g., what suppression measures have been introduced up to time $t$); $\hat{\beta}\left(y_t^o, X_t^o\right)$ is the (asymptotic mean of) the estimated parameters, conditioning on the observable data and information; $T^o$ = 51 is the number of days between March 1st and April 20th, 2020; $SIQRM_t\left(\beta|X_t^o\right)$ is the total number of recorded cases at time $t$ that is projected by the SIQRM model given observable information $X_t^o$ and a set of parameters $\beta$.

Following Green [13], the estimated asymptotic variance-covariance (ESV) matrix of the non-linear least square estimators was calculated using Eq (7):

$$ESV(\beta) = \frac{\sum_{t=0}^{T^o}\left(y_t^o - SIQRM_t\left(\hat{\beta}|X_t^o\right)\right)^2}{T^o} \left[\left(\frac{\partial SIQRM_t\left(\hat{\beta}|X_t^o\right)}{\partial \beta}\right)^{Tr}\left(\frac{\partial SIQRM_t\left(\hat{\beta}|X_t^o\right)}{\partial \beta}\right)\right]^{-1} \qquad (7)$$

where $Tr$ is the transpose matrix operator.

A computational issue arising from the non-linear least squares technique is that it depends on the starting point of each parameter value, and the optimization process may end up with a local rather than the global minimum. Our response was to use a multi-start optimization algorithm. Thus, we repeatedly solved the least-squares optimization process with 1,000 random starting 'guess points' and used the best point to simulate the projection outcomes. These starting 'guess points' were randomized within certain ranges, as suggested by parameter values in the literature. For example, the range for randomizing the initial guess for the basic reproduction rate was 1.4 to 3.25 [14], and the range for the initial guess of the infectious period was between 1 and 14 days [15]. The range for the effectiveness of suppression measures was set between 0 and 1, by the definition of that parameter.

The estimated parameters to forecast the dynamics of the spread of reported COVID-19 was undertaken by randomizing the parameters using their estimated distributions to generate 20,000 simulations with a parallel-processing routine. To test the predictive capacity of the model, we followed a standard procedure that used some part of the available data to fit the model, then projected forward to determine how the model matched the

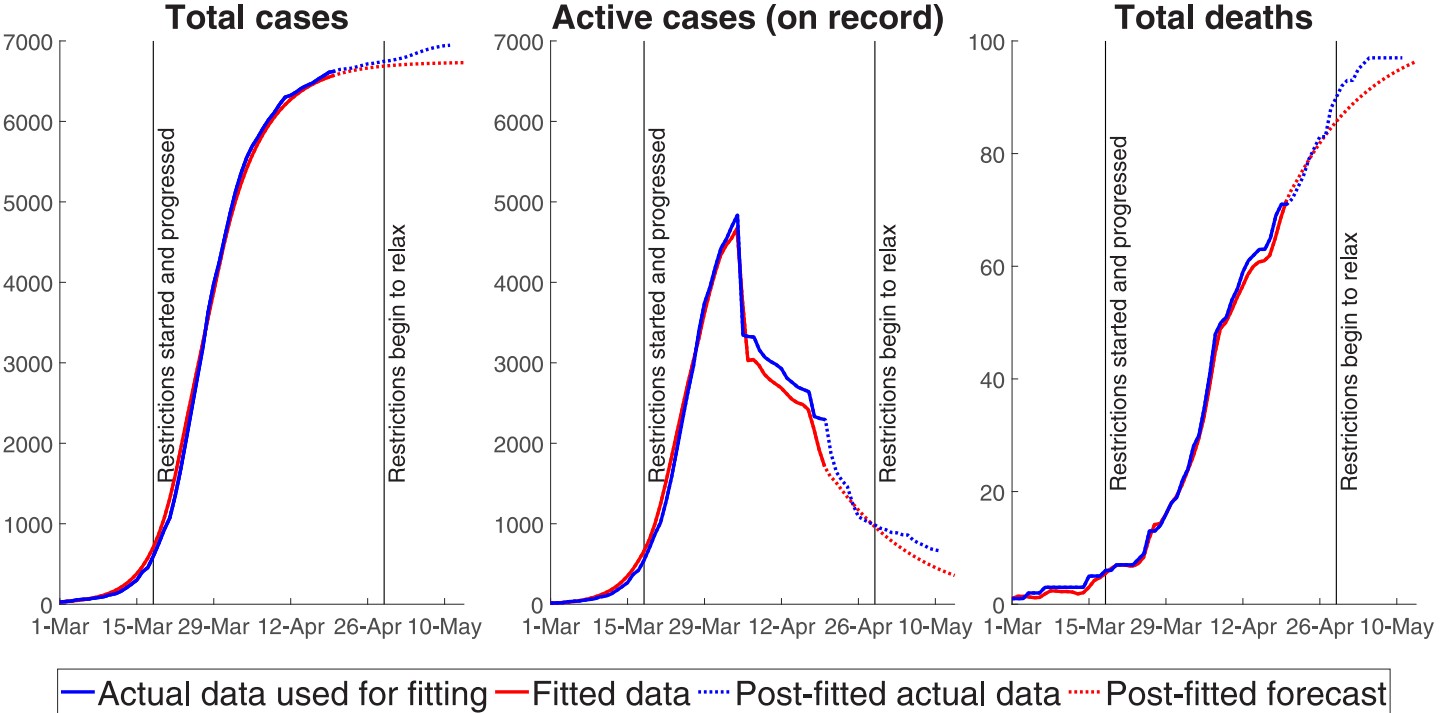

**Fig 3. Projected versus actual data.**

remaining available data. In particular, March 1st (the first recorded death) was considered as time zero, and we used the next 50-day interval (from March 2nd to April 20th, inclusive) to estimate model parameters, or to 'fit the model'. Data from April 21nd were used to determine model 'accuracy'. Suppression measures were introduced to account for the period of time (on average) before symptoms appear and thus when new cases were recorded in the data.

### 3.3. Model fit

Fig 3 compares the mean-value projections generated by our SIQRM model and the actual data (relevant confidence intervals are provided in Fig 4). The three panels compare three observable indicators; namely, the total number of recorded cases, the number of active cases recorded, and the number of fatalities. The red line is model output, the blue line is the actual data, which we used to fit the model to April 20th, and the dotted line is actual post-fitted data. The model outcome for the peak timing of recorded active cases (roughly April 5th) is in the middle panel and closely matches actual data.

The two key estimated epidemiological parameters used to generate Fig 3, along with their 95% confidence intervals, are the (1) average infection period $T^i$ in days, 7.00 [6.85–7.15] and (2) the basic reproduction rate $R_0$, 2.48 [2.45–2.51]. The effectiveness of the three policy measures, estimated as percentage reductions in transmissions compared to the unmitigated spread scenario or, in effect, $(1 - \epsilon)$ in the SIQRM model, are: (3) bans on non-essential gatherings 92.19 [83.71–100], (4) non-essential business shutdown 38.14 [35.50–40.77] and (5) maximum two-person gatherings 3.5 [2.97–4.04]. A higher value of the policy parameter $\epsilon$, or a lower $(1 - \epsilon)$, means a greater effect on reducing the rate of infection. Bans on non-essential

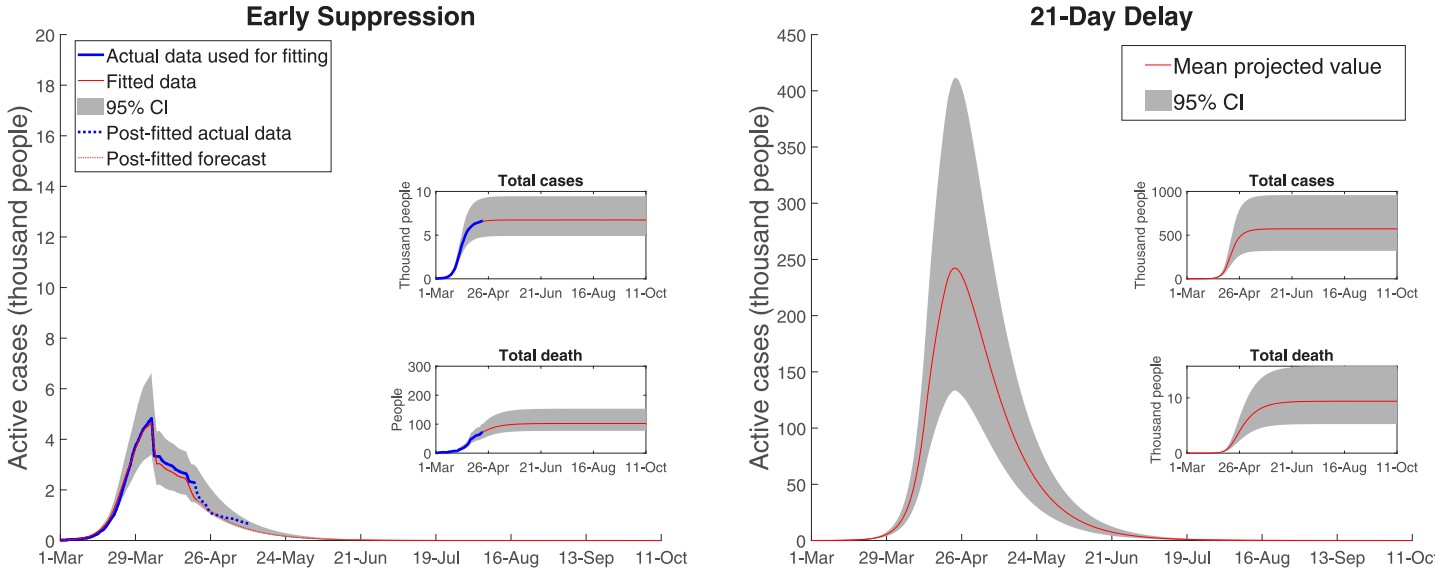

**Fig 4. Australian COVID-19 dynamics: Early and delayed (21 days) suppression.**

gathering, for example, reduced transmission by roughly 8% with the most important mandated suppression measure coming from the limits on the number of people in a gathering.

Our results represent the impacts of a combination of suppression measures, especially non-essential business shut down and limits on gathering, noting that these measures likely reinforce each other. Combined with voluntary social distancing and home isolation, the three suppression measures decreased the extent of the transmission by an estimated (roughly) 96.5%. Our estimates are consistent with surveys by the Australian Bureau of Statistics [16] in the first week of April showing that 98 per cent of the population was practising social distancing and 88 per cent avoided public spaces and events.

There are two additional points to make. First, note that the significant drop in active cases at the beginning of April (and another drop in the third week of April) results from a spike in the number of cases which were classified as 'recovered'—not all of which are predictable by epidemiological parameters alone because how a case was classified as recovered could depend on administrative and reporting procedures. For this reason, during the period where actual data was used to estimate the epidemiological model (i.e., March 1st to April 20th, 2020), we combine the fitted total number of cases (using epidemiological parameters only) and the daily recovery classification rate calculated from the actual data to estimate the number of cases which were reported as recovered. To test the prediction capacity or the accuracy of the epidemiological model, we used a period where the data was 'new' to the model (i.e., April 21st–May 11th, 2020), and in this testing situation, the number of 'recovered' cases was predicted by combining the epidemiologically predicted total number of cases with the average number of sick days of patients who recover, i.e., approximately 18.5 days [5].

Second, note that it is not just the mandated suppression measures that alters transmission, but how the people perceive them and change their social interactions. It is also the case that gathering bans and business shutdowns are highly correlated and subject to uncertainty. We do not account for either in the numerical calculations, given the nature of the compartment model, but we do pick up the point of the impacts of various social distancing outcomes in a related paper [17].

## 3.4. Health care facilities

The requirements for health care facilities (e.g., hospitalization, mechanical, and non-invasive ventilators) are approximated via the following equations:

$$H = h \times Q \tag{8}$$

$$ICU = icu \times Q \tag{9}$$

$$V = v \times Q \tag{10}$$

where $H$ is the number of hospitalized patients and $h$ is the hospitalization ratio (to the number of active cases). We used an estimate from the Australian Department of Health [18] and Garg et al. [19] to specify $h$ = 4.6%; $ICU$ is the number of patients who require an ICU bed and $icu$ is the ratio of those patients to the number of active cases [20]. We specify $icu$ = 1.5% using the estimate of Garg et al. [19] and Fox et al. [21]; $V$ is the number of patients who require a ventilator, and $v$ is the ratio of those patients to the number of active cases. The value of $v$ is estimated to be 1.43% [22].

It is important to note that the unmitigated spread scenario assumes no capacity constraints in hospital beds or ICU units, despite the large increase in infections. This is clearly not realistic. At some point the capacity of the medical system is breached as indicated in the results below, and a lockdown or some severe government response to limit infections would likely ensue. We don't account for this case. In this sense, a comparison of mandated early to delayed suppression is more practical and relevant.

To determine impacts on patient welfare, we used both a Value of a Statistical Life Year (VSLY) measure and an age-adjusted Value of Statistical Life (a-VSL). Note that we only have life expectancy for a given age as it relates to average life expectancy. Throughout, we follow the guidelines for health cost-effectiveness measures, section 3A, contained in [23].

In the first instance, we estimated welfare losses as the difference between normal life expectancy in Australia and average age at the time of death from COVID-19:

$$L = VSLY(t^s \times R + M \times ED) \tag{11}$$

where VSLY is an estimate of the value society places on a year of life, in principle measured by estimating the marginal value or 'willingness to pay' (WTP) to reduce the risk of death. We use an updated VSLY [24], as applied by the Australian Government for public decision-making, of \$213,000, independent of age [25]. $ED$ in Eq (11) is the difference between normal life expectancy and the average age at death of patients who do not recover. We estimated $ED$ = 6.9 years, using the average life expectancy in Australia of 82.5 years [10, 26], and estimate the average age at death from COVID-19 at 75.6 years [5]. The value $t^s$ is the average number of sick days of patients who recover, specified at 18.5 days on average [5].

The estimates in Abelson [24] are drawn from an extensive meta-analysis of prior VSLY and VSL estimates and includes an overall discussion on the major methods of valuation and empirical results for values of life, health and safety in Australia. It also suggests adjustments for those 70 years and older, although we rely instead on Alberini et al. [27] as more technically robust. As usual in cost-benefit studies, Abelson [24] adopts an average WTP value for life, adjusted for age, and thus the VSL is generally held constant regardless of the income of any social group either at any point in time or over time. Most importantly, as conventional, it assumes as a starting point the life of an adult of 40 years of age, likely to live for another 40 years, with again an adjustment for those over 70 years of age.

To avoid a problem with averaging and truncated values of VSLY estimates at the 82.5 average-age lifespan, we use a measure of VSL adjusted by age to account for the fact that most of the deaths in Australia from COVID-19 are among the elderly. Of the 103 fatalities as of 31st May, only 5 were less than 60 years of age, 10 were in their 60s, 34 in their 70s, 34 in their 80s and 20 in their 90s. The VSL was obtained from government estimates and given by $4.9 million [25]. We scaled this Australian VSL by .70 for those over 70 years of age, following Alberini et al. [27], who obtained age-adjusted WTP measures for reductions in mortality risk using contingent valuation surveys. We assumed that the age distribution is unchanged as the number of deaths varies.

## 4. Results

Fig 4 shows both the effect of the 8-weeks lock-down from early suppression and the effect of an assumed 21-day delay in introducing the full range of mandated suppression measures. Early mandated suppression in Australia was highly effective at both 'flattening and shortening' the curve. Our model closely approximates the actual data and estimates the total number of active cases peaks at (roughly) 4,850 cases, with 100 deaths and 6,650 total cases (for a 95% confidence interval of [4,032–12,082] in case load).

With a 21-day delay mandated suppression strategy, the number of active cases peaks at 241,000 [132,000–405,000] and the number of deaths increases to 9,074 [5300–15,360]. The period over which severe suppression measures is extended runs until the third week of July, where there are roughly only 500 active cases remaining. With a 14-day delay the number of fatalities is 2,144 [1,312–3,334] and with a 28-day delay the number of fatalities is 35,374 [19,850–59,070].

Fig 5 shows the case of unmitigated spread, the extreme counterfactual. The total number of cases is nearly 16 million, with a peak of 5.7 million active cases. Fatalities without control, are roughly 260,000 (assuming Australia's current reported mortality rate or reported deaths as a fraction of reported cases). Noting the difference in scale in the two panels in Fig 5, with unmitigated spread, the projected number of active cases still exceeds 2,100 in August 2020 and COVID-19 is not effectively suppressed until late December 2020. Using the lower fatality ratio of 0.7%, as per in Verity et al. [12], results in roughly 112,000 deaths with the unmitigated spread scenario. We include this case, along with the Australian reported fatality rate, in the economic estimates to follow because although the Australian fatality rate on recorded cases predicts well for the lock-down and delayed suppression cases (with their limited horizons), a different fatality rate may apply for the (out of sample) unmitigated spread case.

We have not examined or modelled excess mortality [28] compared to the previous year in Australia, but have a related paper which estimates the true (population) infection rate based on the confirmed number of cases obtained through RNA viral testing [11].

### 4.1. Public health outcomes

Using the results represented in Figs 4 and 5, we projected the demand for health care facilities for early and delayed suppression measures and unmitigated spread. With early mandated suppression, the number of hospital patients is 220 at its peak, with a 95% confidence interval of [130–380]. The peak number of ICU beds and ventilators are, respectively, about 80 and 70 with early mandated suppression. With unmitigated spread, assuming no capacity limits, the number of hospitalized patients peaks at more than 260,000, of which more than 80,000 would need to be placed in an ICU. This ICU use estimate is nearly 40 times higher than the capacity of Australia's health care system, with only 2,229–2,378 ICU beds [29, 30].

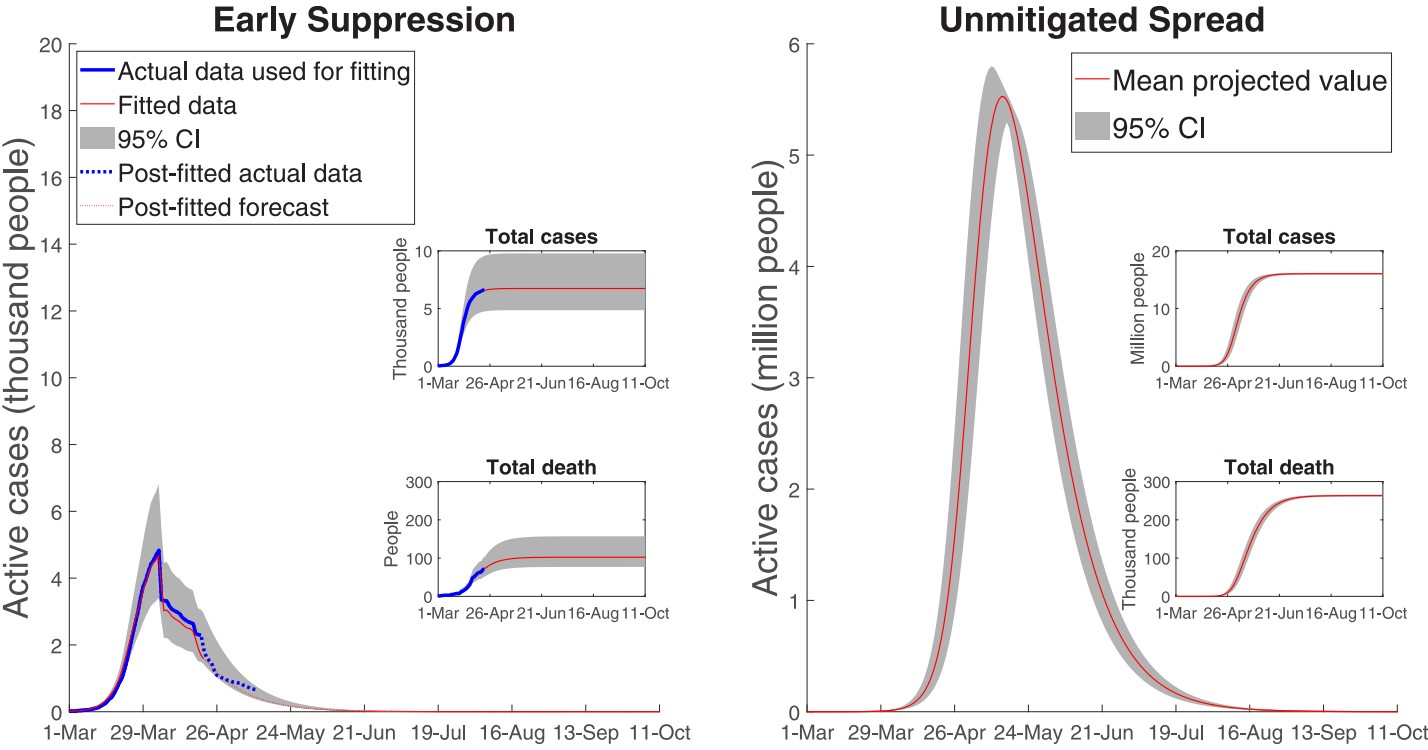

**Fig 5. Australian COVID-19 dynamics: Early and unmitigated spread.**

With unmitigated spread, roughly 70,000 patients would require ventilators at the peak, while there are only around 4,820 machines capable of delivering invasive mechanical ventilation [30]. For 21-day delayed suppression, hospital beds peak at 11,100 [6,000–18,640], and ICUs at 3,585 [1,980–6,075], both also exceeding the capacity of the Australian medical care system.

## 4.2. Welfare losses from Covid19 patients

Tables 1 and 2 provide welfare loss estimates of COVID-19 patients using an Australian VSL of $4.9 million, adjusted by 0.70 for those over 70 years of age, and a VSLY of $231,000 AUD. The VSL estimates in Table 1 vary substantially depending on the scenario. Welfare damages are minimized with early suppression. For the case of a fatality rate of 0.7%, the damage would be $401.6 billion.

Table 2, with the VSLY measure, shows that, with mandated early suppression, the total welfare loss is approximately $229 million AUD with a 95% confidence interval of [$170.4–$335] million AUD. Around 30% of this total cost is attributed to those who recover and around 70% of the costs are from fatalities. For the unmitigated spread scenario, the losses are very large, amounting to $572.8 billion [567.7–577.5], or roughly 2,500 times the loss of early

**Table 1. A-VSL mortality estimates by scenario ($ billion AUD).**

| Early suppression | 14 Day Delay | 21 Day Delay | 28 Day Delay | Unmitigated Spread |
|---|---|---|---|---|
| 0.37 | 7.9 | 34.1 | 129.5 | 956.2 |
| (0.28–0.56) | (4.7–13.1) | (19.1–57.1) | (69.5–221.2) | (947.5–964.6) |

Table 2. Welfare losses of Covid-19 patients: Early versus unmitigated spread.

| | Early Suppression ($ million) | Unmitigated Spread ($ billion) |
|---|---|---|
| Recovered | 80.8 | 192.1 |
| | [58.5–113.7] | [190.3–193.7] |
| Fatalities | 148.2 | 380.8 |
| | [111.9–221.4] | [337.3–383.8] |
| Total | 229 | 572.8 |
| | [170.4–335] | [567.6–577.5] |

suppression. Losses with the 0.7% fatality rate are $240.6 billion, or 1,048 times larger than early suppression.

## 4.3. Health care costs

Table 3 reports the estimate of hospitalization costs, using average measures of bed costs [31, 32], for mandated early suppression and unmitigated spread, using VSLY. For early suppression, the mean value of the total hospitalization costs is $9.8 million AUD of which $3.9 million is the cost of ICU services and $5.9 million is for occupied hospital beds. For the unmitigated spread scenario, the number of hospitalizations is more than 2,300 times higher and the projected hospitalization costs are at least $23.3 billion AUD, with a 95% confidence interval of $23-$27.3 billion AUD. For the 0.7% fatality rate, total costs are $5.88 billion for the unmitigated spread case, or nearly 1000 times larger. Hospitalization costs do not include pre-hospital costs (e.g., such as visits to the GP or ambulance services) or any costs occurring post-hospitalization (e.g., follow-up expenses, and related health risks from having contracted the virus, which are known to be significant).

## 4.4. Direct economy-wide costs

To measure the direct economy-wide costs from early mandated suppression, we used a survey conducted by ABS of business activity during the period from March 30[th] to April 3[rd], after the March 21[st] shut-down of all non-essential businesses, and again in May [33, 34]. In total, 84 percent of businesses reported that mandated suppression measures affected their activity, with many operating at reduced levels, or not at all. Using the reduction of business activity provided by the ABS across all categories, we apportioned the economic cost per day of suppression measures as $C$ by:

$$C = \frac{1}{365} \sum\nolimits_{t=1}^{n} \sum\nolimits_{j=1}^{m} \phi(i,j) GVA(i,j) \tag{12}$$

Table 3. Estimated hospitalization costs: Early versus unmitigated spread.

| | Early Suppression ($ million) | Unmitigated Spread ($ billion) |
|---|---|---|
| Total Costs | 9.8 | 23.3 |
| | (6–18) | (23–23.7) |
| ICU beds | 3.9 | 9.3 |
| | (2.4–7.2) | (9.2–9.5) |
| Hospital beds | 5.9 | 14 |
| | (3.6–10.8) | (13.8–14.2) |

**Table 4. Cost of control measures ($ million per day AUD).**

| Sector/Region | NSW | VIC | QLD | SA | WA | TAS | NT | ACT | Sum |
|---|---|---|---|---|---|---|---|---|---|
| Mining | 11.9 | 3.2 | 30.5 | 2.3 | 67.9 | 0.7 | 4.3 | 0.0 | 120.9 |
| Manufacturing | 29.2 | 23.4 | 19.0 | 4.9 | 10.3 | 1.8 | 1.1 | 0.4 | 89.9 |
| Electricity, gas, water, waste | 6.9 | 4.5 | 5.3 | 1.8 | 2.6 | 0.9 | 0.3 | 0.3 | 22.5 |
| Construction | 31.7 | 26.2 | 19.3 | 5.7 | 14.4 | 1.5 | 3.3 | 1.8 | 103.9 |
| Wholesale trade | 12.0 | 9.6 | 7.6 | 2.3 | 3.5 | 0.5 | 0.2 | 0.3 | 36.0 |
| Retail trade | 8.4 | 7.0 | 5.2 | 1.8 | 2.7 | 0.6 | 0.3 | 0.5 | 26.5 |
| Accom & food services | 6.5 | 3.9 | 3.0 | 0.9 | 1.6 | 0.3 | 0.1 | 0.7 | 16.9 |
| Transport, postal, warehouse | 10.5 | 9.8 | 8.4 | 2.5 | 6.8 | 0.9 | 0.7 | 0.5 | 40.2 |
| Infor media & telecom | 10.3 | 6.6 | 3.4 | 1.5 | 1.9 | 0.4 | 0.2 | 0.4 | 24.7 |
| Financial & insurance | 45.0 | 28.0 | 13.3 | 4.9 | 6.7 | 1.1 | 0.4 | 0.9 | 100.4 |
| Rental, hiring, real estate | 13.1 | 7.1 | 7.1 | 1.6 | 2.7 | 0.6 | 1.0 | 0.8 | 34.0 |
| Professional, scientific & tech | 34.4 | 24.8 | 13.2 | 4.0 | 8.9 | 0.7 | 0.8 | 2.4 | 89.2 |
| Administrative services | 17.6 | 11.4 | 9.6 | 2.8 | 5.4 | 0.4 | 0.5 | 0.6 | 48.3 |
| Education & training | 26.4 | 21.1 | 14.6 | 5.0 | 7.0 | 1.6 | 0.8 | 4.4 | 81.0 |
| Health care, social assistance | 22.5 | 17.6 | 14.6 | 5.7 | 8.5 | 1.4 | 0.9 | 1.6 | 72.7 |
| Arts & recreation services | 2.7 | 1.8 | 1.1 | 0.3 | 0.5 | 0.1 | 0.1 | 0.4 | 7.0 |
| Other services | 4.5 | 3.7 | 2.8 | 0.8 | 1.6 | 0.2 | 0.1 | 0.2 | 14.0 |
| Sum | 293.6 | 209.6 | 177.8 | 48.8 | 153.2 | 13.8 | 15.1 | 16.3 | **928.2** |

Note: Calculations based on source survey material on business activity from IMF [37], ABS [33, 34], and Claughton et al. [36]

where $\phi(i, j)$ is the loss of gross value added in region $i$, sector $j$ given the suppression measures and $GVA(i, j)$ is the gross value of production in 2019 [35].

The value of $\phi(i, j)$ was estimated as:

$$\phi(i,j) = \frac{Y_e(i,j) - Y_c(i,j)}{Y_e(i,j)} \tag{13}$$

where $Y_e(i, j)$ and $Y_c(i, j)$ are the expected and actual level of business before and after the full set of suppression measures were in place for COVID-19, or $\epsilon = 0.965$ in our SIQRM model, drawn from the ABS surveys [33, 34]. It is important to note in these measures that we adjusted the ABS estimates on mining activity following Claughton et al. [36], and that we account for IMF forecasts [37] of a 6% fall in global economic activity, projected to occur (with or without the Australian lock-down).

The value of $C$ is captured in an aggregated form (across the various ABS tables and adjustments) in Table 4. Total losses are $982.2 million per day, or roughly $6.5 billion per week. In terms of order of magnitude, $982.2 million per day is about a 17% fall in daily GDP (using an annual GDP in 2019 of $1,995 billion AUD). Fig 6 shows the impacts by state and territory in Australia, with NSW the most adversely affected.

## 4.5. Transition costs following a lock-down

Our estimated economic losses from 8 weeks of early mandated suppression measures start from March 30th. These economic losses are the economic cost per day times the number of days under the sequence of the introduced control measures, or roughly $51.98 billion AUD over the eight-week period, including all international tourism (see Table 4). As suppression measures are relaxed, in the week of May 25th by our simulation, the economy does not 'snap back' but transitions to full economic activity over time, taking into account that international

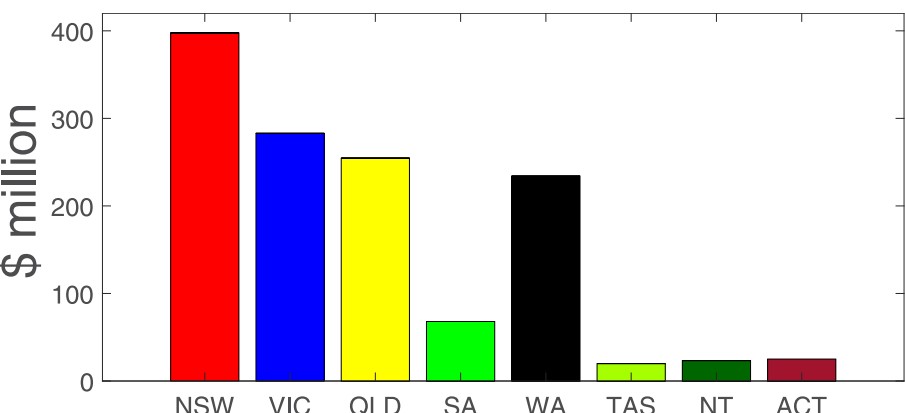

**Fig 6. Cost of COVID control measures by region ($million/day).** Note: NSW = New South Wales; VIC = Victoria; SA = South Australia; WA = Western Australia; TAS = Tasmania; NT = Northern Territory; ACT = Australian Capital Territory.

travel restrictions will still be in place for some time, or at least until 2021 by our calibration. The speed of transition also depends on how quickly government relaxes controls.

In formal terms, the cost of transition is a generalised function of the extent of control measures:

$$C(t) = C(\epsilon)^{1+\delta} = C_{max} \left[ \frac{\epsilon(t)}{1 - \epsilon_{max}} \right]^{1+\delta} \qquad (14)$$

where $C(t)$ is the cost of control measure at time $t$; $\epsilon(t)$ is the policy measure at time $t$, $\epsilon_{max} = .965$ is the most restrictive policy measure, drawn from the SIQRM model, and $\delta$ is a cost-policy parameter. We calibrated Eq (14) with a knowledge of the cost of the lock-down above at $982.2 million per day and the economic costs of controls conditioned on an estimated value of $\epsilon(t) = 0.01$, drawn from our SIQRM model, for the international travel ban, or the 'arrival block' on China, Iran, South Korea and Italy, prior to more severe domestic control measures being put in place. The losses from the arrival block, $C_{ab}$, are given by:

$$C_{ab} = \frac{1}{365} \sum\nolimits_{t=1}^{z} GVGV \ A(i) \qquad (15)$$

where $GVA(i)$ is tourism revenue from region $i$ and $Z$ is number of blocked regions, initially China, Iran, South Korea, and Italy [38]. The total number of passengers from blocked regions is based on the Bureau of Infrastructure, Transport and Regional Economics (BITRE) (the Department of Infrastructure, Regional Development and Cities of Australia) [39]. Based on BITRE [39], the ban on travel from the arrival block reduced total inbound passengers, starting in January, by 17.44%. In 2019, 9.3 million international visitors arrived in Australia with a gross value of tourism of $45.4 billion [38]. In summary, we attribute all the costs of reduction in international tourism to the lockdown. This overstates the costs of lockdown because a decline in international tourism would have occurred even in the absence of stringent social distancing. This is because of Australia's international border controls limit the number of arrivals into Australia and requires supervised quarantine of 14 days. These international border controls have been continuously maintained even when national lockdown ended in May 2020.

**Table 5. Cost of international arrival block.**

| Contents | Unit | Value |
|---|---|---|
| International passengers from banned regions in Jan 2020 | pers | 386,278 |
| Total international passengers in Jan 2020 | pers | 2,214,474 |
| Share of arrival block regions in total passengers | % | 17.13 |
| Share of arrival block regions in total tourism dollars | % | 31.7 |
| GVA of tourism in 2019 | $ billion/year | 45.4 |
| Average loss per day (banned regions) | $ million/day | 39.45 |

Note: Calculations based on source material from BITRE [39] and Tourism of Australia [38]

Results are summarized in Table 5 and using Eq (14) implies:

$$C_{ab} = 39.45 = 928.2 \left[\frac{0.01}{0.965}\right]^{1+\delta} \tag{16}$$

such that

$$1 + \delta = \frac{\ln\left(\frac{39.45}{928.2}\right)}{\ln\left(\frac{0.01}{0.965}\right)} \tag{17}$$

or a value of $\delta = -0.309$.

Using Eq (14), we 'unwind' the government control measures and allow for a return of economic activity until 'full recovery', so that, after transition, the remaining costs are those largely from losses in international tourism from all countries. If the transition time is as short as 1 month, the domestic economy recovers quickly and the losses in GDP are the smallest. If it takes longer for the economy to recover, modelled as a slower fall in $\epsilon$ in our formulation, then depending on the period of time over which the transition occurs, economy losses are larger.

## 4.6. Public health and economy costs combined

Table 6 provides a summary of the total economic losses for four different scenarios, by months of transmission for 1, 2, 3 and 4 months duration, drawing on Australian Bureau of Statistics survey data [33, 34] on the cost of the 8-weeks lock-down itself, which we estimate at $51.98 billion. There are two issues to consider: how quickly suppression measures are relaxed and the time it takes for the economy to recover. With early mandated suppression, lock-down measures can be removed more quickly.

We assume that the cost of transition is roughly half the monthly losses in lock-down in the first month, with costs of transition convex in recovery time in the remaining months. We also note that the cost of the lock-down itself already includes losses in international tourism for all countries, which continue through transition. Given a convex cost of recovery, our simulation gives a range of annualized losses in GDP of early suppression from 3.33% to 6.04% (see Table 6). With no suppression measures, the welfare and hospitalization costs as a percentage of annual GDP range from 13.1% to 47.9%, depending on the welfare measure.

## 5. Discussion

Whether early mandated suppression results in a higher overall economic cost than delayed or no suppression measures requires specification of the severity and duration of the lock-down

**Table 6. Direct economic costs with early suppression in $ billions and GDP loss, and health and welfare losses.**

| | Recovery (months) | Costs ($ billion AUD) | | | Economy Costs (% annual GDP) |
|---|---|---|---|---|---|
| | | Lock-down | Recovery | Total | Annual Loss GDP (%) |
| | | Early Suppression Measures for 8 Weeks from March 30th | | | |
| Transition 1 | 1 | 51.98 | 14.39 | 66.37 | 3.33 |
| Transition 2 | 2 | 51.98 | 30.39 | 82.37 | 4.13 |
| Transition 3 | 3 | 51.98 | 48.27 | 100.26 | 5.03 |
| Transition 4 | 4 | 51.98 | 68.59 | 120.57 | 6.04 |
| | | Welfare Losses, Hospitalization Costs and Fatality Equivalents of Unmitigated Spread | | | |
| | | Welfare | Hospital | Total | Annual Loss GDP (%) |
| VSLY | | 572.8 | 23.3 | 596.1 | 29.8 |
| VSLY* | | 240 | 23.3 | 263.3 | 13.1 |
| A-VSL | | 956.2 | | 956.2 | 47.9 |
| A-VSL* | | 401.6 | | 401.6 | 20.1 |
| Fatality equivalent at %GDP** | | 30,491 (3.3%) | 37,816 (4.13%) | 46,057 (5.03%) | 55,305 (6.04%) |
| Fatality equivalent at %GDP* | | 12,808 (3.3%) | 15,882 (4.13%) | 19,343 (5.03%) | 23,228 (6.04%) |

*VSLY, A-VSL and fatality equivalent measures using the fatality ratio in Verity et al. [12].

**Fatality equivalent is the VSLY-measured number of fatalities under the unmitigated spread scenario that equals the direct economy cost associated with an early 8-weeks lock-down (early suppression) for each % GDP loss (3.33, 4.13, 5.03, and 6.04). N.B. The estimated early suppression model fatalities are 100.

and reliable estimates in relation to key public health parameters. Using a fit-for-purpose SIQRM model for Australia, we estimated the number of cases (cumulative and active), fatalities, hospitalization costs, welfare losses for COVID-19 patients and the loss in economic activity under early suppression and various delays (14, 21 and 28 days), including the case of unmitigated spread. Total estimated hospitalization costs, welfare losses and number of deaths range from more than 1,000 to 4,000 times larger with unmitigated spread compared to early suppression. Delays in suppression (14 to 28 days, or more) provide no economic gain, but increase fatalities and also lengthen the period over which suppression measures are required before active cases fall below 500. This is not to say that the economic cost of the suppression does not depend on the timing, only that fatalities are impacted by the timing of the suppression measures.

Direct economy costs of early mandated suppression depend on how quickly the economy recovers following a lock-down. In the fastest (1 month) assumed recovery after lock-down, the total economy-wide costs are about 3% of GDP. With the slowest assumed recovery (4 months), the total direct economy costs are some 6% of GDP. Current Reserve Bank annualized estimates for the period from January to September of 2020 indicate a fall in Australia-wide GDP of 3.8%. By contrast, the total welfare losses and hospitalization costs range from 12.5% to nearly 48% of GDP with no suppression measures in place. Our results indicate that the total economic costs of unmitigated spread are (roughly) between 4 and over 8 times larger than mandated early suppression (see Table 6).

We provide a number of important caveats to our work. First, our VSLY measure underestimates the welfare losses associated with unmitigated spread. This is because we only have limited information to benchmark on average life expectancy, noting that many of those who die of COVID-19 are older than 82.5 years. In our view, people in this older age bracket would have a both a willingness and ability to pay for additional life. Second, in the case of unmitigated spread, we assume no constraints on hospital and ICU capacity noting that a capacity constraint would likely increase fatalities, and thus welfare losses, should such a constraint be

exceeded. This effect alone may mandate a lockdown at some point. Third, for the unmitigated spread scenario, we do not explicitly model the effects of voluntary physical distancing or a delayed lock-down that could occur with substantial community infections and deaths. We have a related paper that does so [17]. Instead, we estimate the welfare losses (VSLY of fatalities) equivalent to the GDP costs of an 8-weeks lock-down (see Table 6). We find that VSLY welfare losses of fatalities that would be equivalent to GDP losses from an early 8-weeks lock-down results in more than 12,500–30,000 Australian mortalities, depending on the fatality rate–fatalities comparable to countries (e.g., United Kingdom) that implemented delayed lock-downs in 2020. In other words, for early suppression not to be the preferred response to COVID-19, requires that Australia would have to incur more than 12,500–30,000 deaths from unmitigated spread (actual Australian deaths were 102 as of 31st May 2020) than the direct economic costs associated with an early 8-weeks lock-down. Fourth, data limitations preclude us from estimating indirect social costs of unemployment with early or delayed suppression (e.g., additional suicides, domestic violence and alcoholism), but note that many Australian workers have retained employment through a government subsidised and temporary 'Job Keeper' scheme that began on March 30[th], 2020. We have also not accounted for morbidity effects in health outcomes associated with those who recovered form COVID-19.

Notwithstanding our caveats, from both a public health (mortality and morbidity) and direct economic costs (including public health costs) perspective, Australia's mandated early suppression measures that began in March 2020, and which we assume would have continued until the end of May 2020 (suppression measures began to be progressively relaxed from early May), generate a very large economic payoff relative to alternatives of delayed suppression measures or unmitigated spread.

Our findings provide robust evidence that a 'go hard, go early' mandated suppression, at least in a high-income country like Australia, is the preferred approach from both a public health and an economy perspective–a result consistent with the non-technical discussion in Group of Eight Australia [40]. By comparison, some high-income countries adopted a delayed (or much less strict) lock-down with infection rates much higher than when Australia began its lock-down. Our model results suggest that if other high-income countries had imposed effective suppression measures earlier they may have reduced both their public health (including lower fatalities per 1,000 people) and economy costs.

## Supporting information

**S1 Data.**
(XLSX)

## Acknowledgments

The authors are grateful for helpful comments and suggestions provided by Emily Banks, John Baumgartner, Nathaniel Bloomfield, John Parslow, and Andrew Robinson on earlier versions of this manuscript.

## Author Contributions

**Conceptualization:** Tom Kompas, R. Quentin Grafton.

**Data curation:** Tuong Nhu Che.

**Formal analysis:** Tom Kompas, R. Quentin Grafton, Tuong Nhu Che, Long Chu, James Camac.

**Investigation:** James Camac.

**Methodology:** Tom Kompas, Long Chu.

**Validation:** Tom Kompas, Long Chu.

**Writing – original draft:** Tom Kompas, R. Quentin Grafton, Tuong Nhu Che, Long Chu.

**Writing – review & editing:** Tom Kompas, R. Quentin Grafton, James Camac.

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
