## [Decision Letter · Decision Letter 0]

15 Oct 2020

PONE-D-20-18242

Health and Economic Costs of Early, Delayed and No Suppression of COVID-19: The Case of Australia

PLOS ONE

Dear Dr. Kompas,

Thank you for submitting your manuscript to PLOS ONE. After careful consideration, we feel that it has merit but does not fully meet PLOS ONE’s publication criteria as it currently stands. Therefore, we invite you to submit a revised version of the manuscript that addresses the points raised during the review process.

A **rebuttal letter** that responds to **EACH** point raised by the academic editor and reviewer(s). You should upload this letter as a separate file labeled 'Response to Reviewers'.A **marked-up copy** of your manuscript that highlights changes made to the original version. You should upload this as a separate file labeled 'Revised Manuscript with Track Changes'.An **unmarked version** of your revised paper without tracked changes. You should upload this as a separate file labeled 'Manuscript'.

We look forward to receiving your revised manuscript.

Kind regards,

Brecht Devleesschauwer

Academic Editor

PLOS ONE

Additional Editor Comments:

In your revision note, please include EACH of the reviewer comments, provide your reply, and when relevant, include the modified/new text (or motivate why you decided not to modify the text). Note that failure to do so may result in a rejection of the manuscript.

Journal Requirements:

2. Please provide more details on data collection. Specifically, please list the date range for which you obtained data, any inclusion/exclusion criteria, and the categories of data that were extracted.

Reviewers' comments:

Reviewer's Responses to Questions

**Comments to the Author**

1. Is the manuscript technically sound, and do the data support the conclusions?

Reviewer #1: Partly

Reviewer #2: Partly

2. Has the statistical analysis been performed appropriately and rigorously? 

Reviewer #1: Yes

Reviewer #2: I Don't Know

3. Have the authors made all data underlying the findings in their manuscript fully available?

Reviewer #1: Yes

Reviewer #2: No

4. Is the manuscript presented in an intelligible fashion and written in standard English?

Reviewer #1: Yes

Reviewer #2: No

5. Review Comments to the Author

Reviewer #1: The authors tackle very relevant and timely issues on the public health and economic burden and benefits of COVID-19 suppression in Australia. The central question is whether the cure (i.e. the lock-down) may be more ‘costly’ in terms of the economy than delayed or no suppression measures. Therefore, the authors constructed a transmission model to fit the observed incidence by estimating epidemiological and suppression parameters. By delaying or relaxing the suppression, the authors calculate the corresponding burden of disease and health cost and make the comparison with the current situation.

I do have some comments on the manuscript and methods.

The transmission model is based on 5 compartments in terms of S-I-Q-R-M. The observed active cases in Australia show a significant drop at the beginning of April and the model fits this behavior surprisingly well. How did the authors capture this behavior with a homogeneous mixing model? And even more interesting, how come this drop is not present in their scenario analyses on delayed or no suppression measures?

The counter-factual scenario “no suppression” assumes that no voluntary behavioral changes occur. As such, “unmitigated spread” would be a better label for this scenario. In addition, the analysis assumes that cost of hospital and ICU admissions can be linearly extrapolated even if the need outreaches the capacity. If hospital beds or ICU’s are not available, other choices will be made which a different impact on the burden of disease and related costs. As such, the statement that the current suppression has saved XXX lives and XXX dollars is controversial. The comparison with a delayed of relaxed suppression seems more informative and reliable.

Not the suppression measures but how the people conceive them and change their social interactions alters the transmission. In addition, the estimated impact in the SIQRM model of gathering bans and business shutdowns is highly correlated and subject of uncertainty. This uncertainty aspect is missing in the numerical calculations.

The impact calculation of patient welfare is not clear to me. They use both the Value of a Statistical Life Year and an age-adjusted Value of Statistical Life. For the latter, they adopt a government estimate of $4.9 million. How is this calculated and which assumptions are made? This is a crucial parameter in a cost-benefit analysis. Please provide more insights on this.

Australia has clear guidelines for health technology assessments, which are not fully addressed here (https://pbac.pbs.gov.au/information/about-the-guidelines.html) Please make sure all aspects of section 3 “Economic evaluation” are included and reported in the study.

Reviewer #2: Abstract:

• Early (actual): not very clear what you mean with actual

• Second sentence is long (3/4 of the abstract) and hard to read; rephrase

• Clarify the methods used

Introduction

• As of the end of May 2020, the global number of COVID-19 cumulative cases and reported fatalities, respectively, exceed 6 million and 370,000  worldwide?

• Update the data � once accepted use the most recent available data

• (including the President of the United States on 22 March 2020) � not sure if this adds much. Others including scientists, were also posing this question.

• Several language issues: More costly in terms of the economy ; that we simulate continued until the end of May

• The structure is not logical. Research question comes rather early. Background info is limited. Last paragraph should be moved to method section, results are already given in introduction

• 2. COVID-19 in Australia and policy responses � include this in introduction section ; this is the background that we need in order to introduce research question

• Information included under heading 2 can also be shortened � too much detail about progress of pandemic. Main question is, why do we need these analyses?

Methods:

• excluded the below-20-year-old population group � most recent evidence indicated that you should lower the lower bound. Adolescents are also susceptible

• data source not clear

• I do not completely agree with QALY comment, luckily the adjustment for age covers this partially  discuss this issue in more detail. + effect of age adjustement (scenario analyses of different adjustments)

• Life expectancy for given age should be used

• Why not using the GDP as VSLY?

• You only look at PRO of lock down (effect on covid deaths) but not at CON (effect of lock down on general population, delay in usual care, increase in more severe diagnosis after lock down, etc…)

• How did you calculate VSLY for recovered patients?

• Why not looking at (modelled) excess mortality compared to previous year?

Results

• It would be very informative to see the results of individual different measures included in lock down closure of schools, shut down of non-essential business, social distancing

DIscussion

• The discussion lacks comparison with other studies. It is merely a summary of own findings.

• Problem with +82 year olds could be solve by using life expectancy at given age

• What was the influence of time? We see that the number of deaths is now decreasing, even though less measures are in place compared to lock down? How should we look at upcoming months? Go hard, go early? Of a more adapted scenario?

6. PLOS authors have the option to publish the peer review history of their article (what does this mean?). If published, this will include your full peer review and any attached files.

Reviewer #1: No

Reviewer #2: No

---

## [Author Response · Author response to Decision Letter 0]

30 Dec 2020

Responses to Reviewers 

PONE-D-20-18242

Health and Economic Costs of Early, Delayed and No Suppression of COVID-19: The Case of Australia

The authors of this paper sincerely thank the editor and two reviewers for their valuable comments and suggestions. The paper has been greatly improved as a result. Specifically, we have made the following changes in response to the Reviewers’ Comments:

Reviewer #1

The authors tackle very relevant and timely issues on the public health and economic burden and benefits of COVID-19 suppression in Australia. The central question is whether the cure (i.e., the lock-down) may be more ‘costly’ in terms of the economy than delayed or no suppression measures. Therefore, the authors constructed a transmission model to fit the observed incidence by estimating epidemiological and suppression parameters. By delaying or relaxing the suppression, the authors calculate the corresponding burden of disease and health cost and make the comparison with the current situation.

I do have some comments on the manuscript and methods.

Comment #1. The transmission model is based on 5 compartments in terms of S-I-Q-R-M. The observed active cases in Australia show a significant drop at the beginning of April and the model fits this behaviour surprisingly well. How did the authors capture this behaviour with a homogeneous mixing model? And even more interesting, how come this drop is not present in their scenario analyses on delayed or no suppression measures?

Response #1: Thank you. We have revised the paper, including the figure label to clarify this point, with the following in mind. The significant drop in active cases at the beginning of April (and another drop in the third week of April) results from a change in the administrative procedure for how a case was classified as ‘recovered’ during March-April 2020. For example, two active cases appear to have recovered (i.e., no longer contagious) on the same day, but were classified as recovered on different days, and some active cases on record appear to have recovered but were not yet classified due to variable administrative procedures early on and/or different testing rates and capacity during a given week – all of which are not predictable outcomes by our epidemiological parameters alone. For this reason, we estimated the number of cases which were classified recovered by combining the epidemiologically estimated total number of cases and the recorded daily recovery rate whenever actual data were available. As there were a spike in the daily recovery classification rates in the beginning and the third week of April 2020, the number of recovery classifications jumped and the number of active cases on record dropped. We have basically included this text verbatim in the revised paper. 

Comment #2: The counter-factual scenario “no suppression” assumes that no voluntary behavioural changes occur. As such, “unmitigated spread” would be a better label for this scenario. In addition, the analysis assumes that cost of hospital and ICU admissions can be linearly extrapolated even if the need outreaches the capacity. If hospital beds or ICU’s are not available, other choices will be made which a different impact on the burden of disease and related costs. As such, the statement that the current suppression has saved XXX lives and XXX dollars is controversial. The comparison with a delayed of relaxed suppression seems more informative and reliable.

Response #2: We agree, this is a good suggestion. We have changed ‘no suppression’ to ‘unmitigated spread’ throughout, including in the title of the paper. We also highlight your point that ICU admissions and hospitalisations will reach a limit at some point and thus the comparison with ‘delayed suppression’ may be more informative. The text in section 3.4 reads: “It is important to note that the unmitigated spread scenario assumes no capacity constraints in hospital beds or ICU units, despite the large increase in infections. At some point the capacity of the medical system is breached as indicated in the results below, and a lockdown would likely ensue. In this sense, a comparison of early to delayed suppression is more practical and relevant.”

Comment #3. Not the suppression measures but how the people conceive them and change their social interactions alters the transmission. In addition, the estimated impact in the SIQRM model of gathering bans and business shutdowns is highly correlated and subject of uncertainty. This uncertainty aspect is missing in the numerical calculations.

Response #3: Thank you. We acknowledge both points in our revised paper, at the end of section 3.3 and indicate a related work that considers various social distancing outcomes. The text reads: “Second, note that it is not just the mandated suppression measures that alters transmission, but how the people perceive them and change their social interactions. It is also the case that gathering bans and business shutdowns are highly correlated and subject to uncertainty. We do not account for either in the numerical calculations, given the nature of the compartment model, but we do pick up the point of the impacts of various social distancing outcomes in a related paper (Grafton et al. 2020).” 

Comment #4: The impact calculation of patient welfare is not clear to me. They use both the Value of a Statistical Life Year and an age-adjusted Value of Statistical Life. For the latter, they adopt a government estimate of $4.9 million. How is this calculated and which assumptions are made? This is a crucial parameter in a cost-benefit analysis. Please provide more insights on this.

Response #4: (1) Thank you, agreed, this is very important. We use both measures to partly account for age distribution and problems with averaging. VSLY is truncated at an average life expectancy of 82.5 years, which underestimates damages. A-VSL picks up this point and accounts for the fact that most patients who die from COVID in Australia are elderly. (2) We rely on Abelson (2007), the standard reference used by government in Australia to obtain the $4.9 million figure and take greater care to articulate the procedure and assumptions that are adopted here. Please see section 3.4, including the added text: “The estimates in Abelson (2007) are drawn from an extensive meta-analysis of prior VSLY and VSL estimates and includes an overall discussion on the major methods of valuation and empirical results for values of life, health and safety in Australia. It also suggests adjustments for those 70 years and older, although we rely instead on Albertini et al. (2004) as more technically robust. As usual in cost-benefit studies, Abelson (2007) adopts an average WTP value for life, adjusted for age, and thus the VSL is generally held constant regardless of the income of any social group either at any point in time or over time. Most importantly, as conventional, it assumes as a starting point the life of an adult of 40 years of age, likely to live for another 40 years, with again an adjustment for those over 70 years of age.”

Comment #5. Australia has clear guidelines for health technology assessments, which are not fully addressed here (https://pbac.pbs.gov.au/information/about-the-guidelines.html) Please make sure all aspects of section 3 “Economic evaluation” are included and reported in the study.

Response #5: Thank you. As you suggested, we have revised our paper to highlight the link you have checked to make certain that we have captured all aspects of section 3 of the guidelines. In particular, as listed in section 3A of the guidelines (Cost-Effectiveness Analysis), our revised paper includes an overview and rationale of the economic evaluation, and a description of the computational method, and population, including model variables and extrapolation. Our paper also includes and discusses health outcomes, health care resource uses and costs, model validation, results for the base-case (our ‘unmitigated spread’ or ‘delayed suppression’ case, depending on context), economic evaluation, as well as uncertainty analysis.

 

Reviewer #2 

Comment #6: Abstract:

• Early (actual): not very clear what you mean with actual

• Second sentence is long (3/4 of the abstract) and hard to read; rephrase

• Clarify the methods used

Response #6: 

• We have removed the term ‘actual’ from the abstract, it is confusing. Active cases are defined clearly in the paper in any case. 

• Thank you, good point. We have rephrased the second sentence, re-worked the abstract and added detail on the methods used. 

Comment #7: Introduction

• As of the end of May 2020, the global number of COVID-19 cumulative cases and reported fatalities, respectively, exceed 6 million and 370,000  worldwide?

• Update the data � once accepted use the most recent available data

• (including the President of the United States on 22 March 2020) � not sure if this adds much. Others including scientists, were also posing this question.

• Several language issues: More costly in terms of the economy ; that we simulate continued until the end of May

• The structure is not logical. Research question comes rather early. Background info is limited. Last paragraph should be moved to method section, results are already given in introduction

• 2. COVID-19 in Australia and policy responses � include this in introduction section ; this is the background that we need in order to introduce research question

• Information included under heading 2 can also be shortened � too much detail about progress of pandemic. Main question is, why do we need these analyses?

Response #7: 

• Yes, global data available at the time. 

• We are going to retain the data relevant to the May 2020 period since it fits the context of the paper and the comparisons we make. 

• We have removed the statement from the President, rightly so. Thank you. 

• Thank you. We have carefully checked and revised the text. 

• Thank you. We have revised the Introduction and, in particular, removed the results from this section. 

• It appears to us that Section 2 follows logically from the Introduction, so we have left it as is and we prefer to keep all of the detail (on active case numbers and including the timing of the suppression controls) since it provides the needed context. We have shifted some of the material in the Introduction to this section. 

Comment #8: Methods:

• excluded the below-20-year-old population group � most recent evidence indicated that you should lower the lower bound. Adolescents are also susceptible

• data source not clear

• I do not completely agree with QALY comment, luckily the adjustment for age covers this partially  discuss this issue in more detail. + effect of age adjustment (scenario analyses of different adjustments)

• Life expectancy for given age should be used

• Why not using the GDP as VSLY?

• You only look at PRO of lock down (effect on COVID deaths) but not at CON (effect of lock down on general population, delay in usual care, increase in more severe diagnosis after lock down, etc…)

• How did you calculate VSLY for recovered patients?

• Why not looking at (modelled) excess mortality compared to previous year?

Response #8: 

• We have revised this section to indicate that we have not excluded the under 20-year group in our modelling, while still noting throughout the paper that almost all of the fatalities are over 60 years old. 

• We have double-checked to make certain that the references are clear. Data on all compartment categories is official government data, as indicated, mostly drawn from Covid-19-Data (2020). 

• Thank you, we have added further discussion of ‘age adjustments’ to the welfare measures throughout but left the calculations as a .70 reduction for those over 70 years unchanged for the adjusted value of a statistical life, following Abelson (2007). Please also see section 3.4, including the added text: “The estimates in Abelson (2007) are drawn from an extensive meta-analysis of prior VSLY and VSL estimates and includes an overall discussion on the major methods of valuation and empirical results for values of life, health and safety in Australia. It also suggests adjustments for those 70 years and older, although we rely instead on Albertini et al. (2004) as more technically robust. As usual in cost-benefit studies, Abelson (2007) adopts an average WTP value for life, adjusted for age, and thus the VSL is generally held constant regardless of the income of any social group either at any point in time or over time. Most importantly, as conventional, it assumes as a starting point the life of an adult of 40 years of age, likely to live for another 40 years, with again an adjustment for those over 70 years of age.”

• We only have life expectancy for a given age as it relates to average life expectancy. We have clarified this point. 

• We have measures for the fall in GDP but also included VSLY and A-VSL because these are standard measures. 

• We have briefly highlighted the negative impacts of ‘lockdown’, while still noting that many of these impacts were mitigated through government assistance programs in the Discussion section. We have also re-emphasized that morbidity effects from COVID are not included. Unfortunately, there is no available quantitative measures to be able to include in our modelling. 

• Thank you. We measure VSLY only for mortalities and account for hospitalisation and other costs for those patients who die and ultimately recover separately. We have clarified this point. We do account for the average number of sick days of patients who do recover, specified at 18.5 days on average. 

• Thank you. We have provided additional references to acknowledge this point, just before section 4.1. 

Comment #9: Results

• It would be very informative to see the results of individual different measures included in lock down closure of schools, shut down of non-essential business, social distancing.

Response #9: Agreed, unfortunately there is insufficient data for us to include in our modelling. 

Comment #10: Discussion

• The discussion lacks comparison with other studies. It is merely a summary of own findings.

• Problem with +82 year olds could be solve by using life expectancy at given age

• What was the influence of time? We see that the number of deaths is now decreasing, even though less measures are in place compared to lock down? How should we look at upcoming months? Go hard, go early? Of a more adapted scenario?

Response #10: 

• Thank you. To the best of our knowledge there is no published material or pre-print that combines an epidemiological model with the impacts on the economy from various mandated suppression measures (including the case of unmitigated spread) for Australia – save for a non-technical discussion in Group of Eight Australia (2020) and also Grafton et al. (2020) which are cited in the revised manuscript. 

• We only have access to average life expectancy but note your suggestion/qualification in the revised paper. 

• Thank you. We contend (based on our modelling results) that a ‘go hard, go early’ is the preferred public policy choice.

---

## [Decision Letter · Decision Letter 1]

25 Feb 2021

PONE-D-20-18242R1

Health and Economic Costs of Early and Delayed Suppression and the Unmitigated Spread of COVID-19: The Case of Australia

PLOS ONE

Dear Dr. Kompas,

Thank you for submitting your manuscript to PLOS ONE. After careful consideration, we feel that it has merit but does not fully meet PLOS ONE’s publication criteria as it currently stands. Therefore, we invite you to submit a revised version of the manuscript that addresses the points raised during the review process.

We look forward to receiving your revised manuscript.

Kind regards,

Brecht Devleesschauwer

Academic Editor

PLOS ONE

Journal Requirements:

Additional Editor Comments (if provided):

Thank you for addressing the reviewer comments. Reviewer #1 raised some further remarks, which can be addressed in a final, minor revision round. In your response to Reviewer #2, I had also noted that you had not always provided an adequate response -- for each comment, please make sure to refer us to changes made in the manuscript, or indicate why you decided not to change the manuscript. Only providing a reply in the revision note is not sufficient, because readers of the article will not have access to these notes.

Reviewers' comments:

Reviewer's Responses to Questions

**Comments to the Author**

1. If the authors have adequately addressed your comments raised in a previous round of review and you feel that this manuscript is now acceptable for publication, you may indicate that here to bypass the “Comments to the Author” section, enter your conflict of interest statement in the “Confidential to Editor” section, and submit your "Accept" recommendation.

Reviewer #1: (No Response)

2. Is the manuscript technically sound, and do the data support the conclusions?

Reviewer #1: Yes

3. Has the statistical analysis been performed appropriately and rigorously? 

Reviewer #1: Yes

4. Have the authors made all data underlying the findings in their manuscript fully available?

Reviewer #1: Yes

5. Is the manuscript presented in an intelligible fashion and written in standard English?

Reviewer #1: Yes

6. Review Comments to the Author

Reviewer #1: The authors improved their manuscript and resolved many issues from both reviewers. Thank you for your gratitude and by taking our feedback seriously. The current message is more nuanced and thus informative, in my opinion. I still have a few comments.

The authors replied on my question (#1) why the reported cases dropped at the beginning of April. The added text (line 175-180) is rather long and difficult to read. However, changes in reporting strategy and other administrative issues have taken place worldwide and interfere with modelling, though my question was, “how did your SIQRM model capture this sudden drop”? What kind of temporal parameter(s) did you include to capture this observed behavior? How did you inform these parameters? And what consequences does this additional temporal “model intervention” has for making predictions?

The counterfactual scenario on “no suppression” is well explained in the revised manuscript, though the additional note that “a lockdown would likely ensure” seems to undermine the scenario. This is stated at line 206 and 440. In my comment #2, I referred to “a total breakdown of the medical system will be even more deadly than the estimates, since there are not enough e.g. ICU’s to save severe cases and all other care is not possible. The costs are uncertain, since you cannot pay for ICU’s or ventilators that are not there. Many patients will not be admitted to ICU, though you include unlimited ICU costs”. The bottom line of this scenario, is that there is no lockdown.

On line 211, the authors explicitly challenge the usefulness of QALYs for cost-effectiveness comparisons among alternative public health responses. Unless they have a vast amount of literature to backup this claim, I would just mention that the analysis is based on life-years-lost and not on QALYs, in line with the Guidelines.

On line 244, the authors report their scaling of the VSL of 4.9million dollar by 0.7 for people over 70 years of age. This is not clear to me. So, 70-year olds represents only/still 30% of their monetary value to society? Please explain this more for the reader.

The impact of mandated suppression on tourism seems not completely fair. It is sure that the measures in place prevented people from coming to Australia, and this has an economic cost. However, this pandemic also affects outgoing passenger flows in other countries. Even without lockdown, there will be economic loses for tourism. Hence, accounting the full economic loss for tourism to the lockdown requires some remarks.

Line 421 states that “delays in suppression provide no economic gain, but increased fatalities and …). This gives the impression that the economic cost of the suppression does not depend on the timing. Only the fatalities are impacted by the timing of the suppression measures. Would it be possible to elaborate more on this?

The following statement in the abstract and partly repeated in the discussion (line 449) is not clear to me:

"We also find that using an equivalent VSLY welfare loss from fatalities to estimated GDP losses, drawn from survey data and our own estimates of the impact of suppression measures on the economy, means that for early suppression not to be the preferred strategy requires that Australians prefer more than 12,500 - 30,000 deaths, depending on the fatality rate, to the economy costs of early mandated suppression. "

This reads like Australians have to choose between early suppression or 12500+ deaths. This seems not the take-home message from this manuscript. What about the nuance between delayed suppression? In addition, this assumes a fixed willingness-to-pay which is rather a complex concept and a subject for discussion.

Minor comments

- “Active cases” is rather uncommon. The number of active cases over time can be indicated by the prevalence.

- Line 60, how did the authors calculated the daily growth rate?

- The manuscript contains ‘Error! Reference source not found”

- Line 325: what do the authors mean with “Some 84 percent of businesses…” ?

- Line 344 misses the Figure number

- Line 436-437 is not clear to me.

- Line 432, what do the authors mean with “the spread of COVID-19” ?

- Line 471, please rephrase “public health costs and economy costs”.

- In the equations, the letter “T” is re-used in different equations. This is confusing.

- Line 105, R0 is defined as the basic reproduction number and not the “initial” reproduction number.

- The manuscript does not include any comment or remark that this model is an abstraction of the incubation, symptomatic, pre-symptomatic and asymptomatic infection stages. The model does not include any time lag between infection and infectiousness, which has impact on the timing of the predicted epidemic. The model structure is clearly stated and fits the purpose, though a remark on this issue would put the analyses in perspective.

7. PLOS authors have the option to publish the peer review history of their article (what does this mean?). If published, this will include your full peer review and any attached files.

Reviewer #1: No

---

## [Author Response · Author response to Decision Letter 1]

30 Apr 2021

Response to Reviewer(s) 

Thanks again to Reviewer #1 for providing very helpful comments. We greatly appreciate the time she/he has spent with our paper! Our responses are below. 

Reviewer #1: The authors improved their manuscript and resolved many issues from both reviewers. Thank you for your gratitude and by taking our feedback seriously. The current message is more nuanced and thus informative, in my opinion. I still have a few comments.

1. The authors replied on my question (#1) why the reported cases dropped at the beginning of April. The added text (line 175-180) is rather long and difficult to read. However, changes in reporting strategy and other administrative issues have taken place worldwide and interfere with modelling, though my question was, “how did your SIQRM model capture this sudden drop”? What kind of temporal parameter(s) did you include to capture this observed behavior? How did you inform these parameters? And what consequences does this additional temporal “model intervention” has for making predictions?

Response: Thank you for this point. We have clarified the text to read: ‘There are two additional points to make. First, note that the significant drop in active cases at the beginning of April (and another drop in the third week of April) results from a spike in the number of cases which were classified as ‘recovered’-- not all of which are predictable by epidemiological parameters alone because how a case was classified as recovered could depend on administrative and reporting procedures. For this reason, during the period where actual data was used to estimate the epidemiological model (i.e., March 1st to April 20th, 2020), we combine the fitted total number of cases (using epidemiological parameters only) and the daily recovery classification rate calculated from the actual data to estimate the number of cases which were reported as recovered. To test the prediction capacity or the accuracy of the epidemiological model, we used a period where the data was ‘new’ to the model (i.e., April 21st – May 11th, 2020), and in this testing situation, the number of ‘recovered’ cases was predicted by combining the epidemiologically predicted total number of cases with the average number of sick days of patients who recover, i.e., approximately 18.5 days (Covid19-Data, 2020).’ 

2. The counterfactual scenario on “no suppression” is well explained in the revised manuscript, though the additional note that “a lockdown would likely ensure” seems to undermine the scenario. This is stated at line 206 and 440. In my comment #2, I referred to “a total breakdown of the medical system will be even more deadly than the estimates, since there are not enough e.g. ICU’s to save severe cases and all other care is not possible. The costs are uncertain, since you cannot pay for ICU’s or ventilators that are not there. Many patients will not be admitted to ICU, though you include unlimited ICU costs”. The bottom line of this scenario, is that there is no lockdown.

Response: Understood, thank you. Our only point is that if the capacity of the medial system is exceeded the government position would indeed be to enforce a severe lockdown. Our counterfactual ignores this, of course. We have adjusted the text (near line 206) to clarify the point. 

3. On line 211, the authors explicitly challenge the usefulness of QALYs for cost-effectiveness comparisons among alternative public health responses. Unless they have a vast amount of literature to backup this claim, I would just mention that the analysis is based on life-years-lost and not on QALYs, in line with the Guidelines. On line 244, the authors report their scaling of the VSL of 4.9million dollar by 0.7 for people over 70 years of age. This is not clear to me. So, 70-year olds represents only/still 30% of their monetary value to society? Please explain this more for the reader.

Response: Thank you. We have removed this part of the discussion on QALYs.

Response: We have kept the text as is, although certainly acknowledge that the value of a statistical life can be calculated in different ways and, depending on the method used, may vary with a person's age. If based on future earnings, then persons who are young adults may have a higher relative VSL to an older demographic. If based on ability and willingness to pay (WTP) measures, middle-aged or older persons may have a higher relative VSL to a younger demographic. We followed the findings of Alberini et al. (2004) who found weak support that WTP declines with age, and only for older respondents (aged 70 and above) in their surveys of two populations (one in Canada and one in the US). They found that their Canadian survey respondents over the age of 70 were willing to pay about one-third less than their younger counterparts to reduce their risk of dying by 5 in 1000 over the next 10 years. Thus, we adopted the heuristic to reduce VSL of persons over 70 years of age by 0.30 noting that a similar heuristic (0.75 X VSL for persons 65 years and older) has previously been employed by Health Canada.

4. The impact of mandated suppression on tourism seems not completely fair. It is sure that the measures in place prevented people from coming to Australia, and this has an economic cost. However, this pandemic also affects outgoing passenger flows in other countries. Even without lockdown, there will be economic loses for tourism. Hence, accounting the full economic loss for tourism to the lockdown requires some remarks.

Response: We agree. The decline in tourism to Australia is primarily a result of the supervised 14-days quarantine on all arrivals, the cost of which must be paid by those arriving in Australia. This is approximately $3,500 per person. Supervised quarantine requires a large number of quarantine workers and suitable rooms to isolate travellers. Resourcing constraints, particularly a sufficient number of beds in appropriate hotels, has also meant the that the number of people permitted to come to Australia is about 20,000 per month. (see https://www.abs.gov.au/statistics/industry/tourism-and-transport/overseas-arrivals-and-departures-australia/latest-release) while pre-COVID in 2019 arrivals were, on average, 700,000 per month. Further, these constraints on arrivals continue despite the fact that the national lockdown ended in May 2020. In sum, attributing the full loss in tourism to the lockdown, as we have done, overestimates the lockdown cost. Thus, in the revised version we state: 'In summary, we attribute all the costs of reduction in international tourism to the lockdown. This overstates the costs of lockdown because a decline in international tourism would have occurred even in the absence of stringent social distancing. This is because of Australia's international border controls limit the number of arrivals into Australia and requires supervised quarantine of 14 days. These international border controls have been continuously maintained even when national lockdown ended in May 2020.' (line 376)

5. Line 421 states that “delays in suppression provide no economic gain, but increased fatalities and …). This gives the impression that the economic cost of the suppression does not depend on the timing. Only the fatalities are impacted by the timing of the suppression measures. Would it be possible to elaborate more on this?

Response: Thanks for this. We have added your qualification. 

6. The following statement in the abstract and partly repeated in the discussion (line 449) is not clear to me: "We also find that using an equivalent VSLY welfare loss from fatalities to estimated GDP losses, drawn from survey data and our own estimates of the impact of suppression measures on the economy, means that for early suppression not to be the preferred strategy requires that Australians prefer more than 12,500 - 30,000 deaths, depending on the fatality rate, to the economy costs of early mandated suppression.” This reads like Australians have to choose between early suppression or 12500+ deaths. This seems not the take-home message from this manuscript. What about the nuance between delayed suppression? In addition, this assumes a fixed willingness-to-pay which is rather a complex concept and a subject for discussion.

Response: Yes, agreed, good point. The use of ‘Australians prefer’ is clearly bad form. We’ve reworded the material (in the abstract and conclusion) to pick this up and the point on unmitigated spread, indicating just model results. The abstract reads ‘ … from survey data and our own estimates of the impact of suppression measures on the economy, means that for early suppression not to be the preferred strategy requires that Australia would have to incur more than 12,500 - 30,000 deaths, depending on the fatality rate with unmitigated spread, to the economy costs of early mandated suppression. We also find that early rather than delayed mandated suppression imposes much lower economy and health costs and conclude that in high-income countries, like Australia, a ‘go early, go hard’ strategy to suppress COVID-19 results in the lowest estimated public health and economy costs.’ We have left aside your point of assuming a fixed willingness to pay as indeed too complex in this version of the paper – something for further research. 

7. Minor comments

- “Active cases” is rather uncommon. The number of active cases over time can be indicated by the prevalence.

Response: We have kept it as is, as standard practice in Australia at the time. Active cases are the numbers of individuals identified and in quarantine. 

- Line 60, how did the authors calculated the daily growth rate?

Response: We have clarified this in the text to read: ‘As of 1st June 2020, Australian mandated suppression measures drastically reduced community transmission of the virus and the daily growth rate (the daily increase in the total number of cases over the total number of cases, on a three-day average) declined from around 25%, with 268 new recorded cases on 22nd March, to a daily growth rate of 0.26% and 11 new recorded cases on 30th May 2020 (Covid19-Data, 2020).’ 

- The manuscript contains ‘Error! Reference source not found”

Response: Repaired. It came through this way in the PDF but is fine in the Word document. In any case, we have re-entered the source. 

- Line 325: what do the authors mean with “Some 84 percent of businesses…” ?

Response: It is 84%. We have removed the ‘some’. 

- Line 344 misses the Figure number

Response: Repaired, thanks. 

- Line 432, what do the authors mean with “the spread of COVID-19” ?

Response: Thanks, we have removed this sentence. It wasn’t needed. 

- Line 471, please rephrase “public health costs and economy costs”.

Response: Repaired, thanks. 

- In the equations, the letter “T” is re-used in different equations. This is confusing.

Response: Thank you for this point. We have revised the paper to clarify that T (without the superscript) is the total number of observed cases, and the transpose matrix operator, which was also denoted as T, is now denoted as Tr.

- Line 105, R0 is defined as the basic reproduction number and not the “initial” reproduction number.

Response: We have revised this sentence as suggested, thank you. 

- The manuscript does not include any comment or remark that this model is an abstraction of the incubation, symptomatic, pre-symptomatic and asymptomatic infection stages. The model does not include any time lag between infection and infectiousness, which has impact on the timing of the predicted epidemic. The model structure is clearly stated and fits the purpose, though a remark on this issue would put the analyses in perspective.

Response: Thank you. We have included a revised version of your qualifier in the technical section (near line 101), which reads as ‘It is important to note that the model, although fit for purpose, is an abstraction of the relevant infection stages and does not include any explicit time lag between infection and infectiousness, which has impact on the practical timing of the predicted epidemic in Australia.’

---

## [Editor Report · Decision Letter 2]

17 May 2021

Health and Economic Costs of Early and Delayed Suppression and the Unmitigated Spread of COVID-19: The Case of Australia

PONE-D-20-18242R2

Dear Dr. Kompas,

We’re pleased to inform you that your manuscript has been judged scientifically suitable for publication and will be formally accepted for publication once it meets all outstanding technical requirements.

Kind regards,

Brecht Devleesschauwer

Academic Editor

PLOS ONE
---

## [Editor Report · Acceptance letter]

25 May 2021

PONE-D-20-18242R2 

Health and Economic Costs of Early and Delayed Suppression and the Unmitigated Spread of COVID-19: The Case of Australia 

Dear Dr. Kompas:

I'm pleased to inform you that your manuscript has been deemed suitable for publication in PLOS ONE. Congratulations! Your manuscript is now with our production department. 

Kind regards, 

on behalf of

Prof. Dr. Brecht Devleesschauwer 

Academic Editor

PLOS ONE